

# DNA barcoding unveils a high diversity of caddisflies (Trichoptera) in the Mount Halimun Salak National Park (West Java; Indonesia)

Isabel C. Kilian[1], Marianne Espeland[1], Wolfram Mey[2], Daisy Wowor[3], Renny K. Hadiaty[3,†] Thomas von Rintelen[2] and Fabian Herder[1]

[1] Museum Koenig, Leibniz Institute for the Analysis of Biodiversity Change, Bonn, Germany
[2] Museum für Naturkunde, Leibniz Institute for Evolution and Biodiversity Science, Berlin, Germany
[3] Museum Zoologicum Bogoriense, Research Center for Biosystematics and Evolution, National Research and Innovation Agency, Cibinong, Indonesia
[†] Deceased author.

## ABSTRACT

**Background:** Trichoptera are one of the most diverse groups of freshwater insects worldwide and one of the main bioindicators for freshwater quality. However, in many areas, caddisflies remain understudied due to lack of taxonomic expertise. Meanwhile, globally increasing anthropogenic stress on freshwater streams also threatens Trichoptera diversity.
**Methods:** To assess the Trichoptera diversity of the area within and around the Mount Halimun Salak National Park (MHSNP or Taman Nasional Gunung Halimun Salak) in West Java (Indonesia), we conducted a molecular-morphological study on Trichoptera diversity using larvae from a benthic survey and adults from hand-netting. In addition to morphological identification, we applied four different molecular taxon delimitation approaches (Generalized Mixed Yule Coalescent, Bayesian Poisson Tree Processes, Automatic Barcode Gap Discovery and Assemble Species by Automatic Partitioning) based on DNA barcoding of Cytochrome-C-Oxidase I (COI).
**Results:** The molecular delimitation detected 72 to 81 Operational Taxonomic Units (OTU). Only five OTUs could be identified to species level by comparing sequences against the BOLD database using BLAST, and four more to the genus level. Adults and larvae could be successfully associated in 18 cases across six families. The high diversity of Trichoptera in this area highlights their potential as bioindicators for water quality assessment.
**Conclusions:** This study provides an example of how molecular approaches can benefit the exploration of hidden diversity in unexplored areas and can be a valuable tool to link life stages. However, our study also highlights the need to improve DNA barcode reference libraries of Trichoptera for the Oriental region.

Corresponding author
Isabel C. Kilian,
i.kilian@leibniz-zfmk.de

## INTRODUCTION

Trichoptera (caddisflies), with currently over 16,260 described species, represent one of the largest orders of primarily aquatic insect species worldwide (*Morse, 2020*). In tropical regions, the high Trichoptera diversity has been discussed in the context of a high diversity of larval adaptations to different habitat types (*Mackay & Wiggins, 1979*), and high rates of endemism in mountainous areas (*de Moor & Ivanov, 2008*). Habitat requirements of caddisfly larvae vary significantly among species, which is why they are frequently used as a bioindicator for monitoring water quality (*Ab Hamid & Md Rawi, 2017*; *Bonada et al., 2006*; *Graf et al., 2008*; *Schmidt-Kloiber & Hering, 2015*). However, to use it properly as bioindicators requires the availability of good information about species taxonomy and habitat requirements in the area of interest, which is rarely the case, especially in tropical and remote areas (*Geraci, Al-Saffar & Zhou, 2011*; *Hoppeler et al., 2016*). Moreover, caddisfly taxonomy is largely based on traits of adult males, whereas larval morphology remains unknown for many species (*Zhou, Kjer & Morse, 2007*). A recent estimate predicts that around 13,000 Trichoptera species are awaiting recognition as formal species (*Zhou et al., 2016*), which further complicates their use as bioindicators.

In Indonesia, knowledge of the caddisfly fauna remains very limited. Previous studies have revealed that Java with 146 species is one of the most diverse islands, exceeding considerably Bali (73 species) and Lombok (61 species) (*Malicky, Ivanov & Melnitsky, 2014*). The MHSNP (Mount Halimun Salak National Park) in southern West Java contains one of the last remaining sub-montane forests in this part of the island (*Kahono, 2003*; *Whitten, Soeriaatmadja & Afiff, 1996*). Located near the capital city of Jakarta, it serves as the major water reservoir for this megacity (*Peggie & Harmonis, 2014*), since its rivers and streams provide water also during the dry season. Due to its richness of habitats, ranging from lowland and lower montane rain forest to montane forest, this park belongs to the area with the highest biodiversity in Java (*Kahono, 2003*). Still, due to some settlements, agricultural practices, and illegal gold mining in the national park, the anthropogenic pressure on rivers increases especially during the dry season (*Galudra et al., 2005*; *Yoga et al., 2014b*). Studies of freshwater organisms inside the National Park have only been conducted sporadically [*e.g.*, on crustaceans by *Ng & Wowor (2018, 2019)*, dragonflies by *Aswari (2004)*, or on snails by *Heryanto (2001)*]. The only study available on aquatic insect diversity in this area recognized a total of 24 caddisfly species, including 12 families of Trichoptera (*Rizali, Buchori & Triwidodo, 2002*). However, the sampling areas, though located within the park, were highly disturbed. Generally, ecological as well as molecular studies of Indonesian caddisfly fauna are largely missing, which also pertains to DNA barcoding efforts in this area.

DNA barcoding takes advantage of interspecific variability in a 658-base-pair (bp) long part of the mitochondrial gene Cytochrome-C-Oxidase I (COI) and is a valuable standard tool for studying unknown diversity, especially in groups where taxonomic expertise is scarce (*Borisenko, Sones & Hebert, 2009*; *Hebert, Ratnasingham & de Waard, 2003*). It has been used successfully to assess species diversity in understudied areas (*De Araujo et al., 2018*; *Cordero, Sánchez-Ramírez & Currie, 2017*; *Geraci, Al-Saffar & Zhou, 2011*; *Janzen &*

*Hallwachs, 2011*) and provides insights into cryptic species diversity (*Hebert et al., 2004*; *Hüllen et al., 2020*; *Pauls et al., 2010*; *Tyagi et al., 2019*). The method has in many cases facilitated the association between life stages without the time-consuming rearing of specimens (*Ahrens, Monaghan & Vogler, 2007*; *Gattolliat & Monaghan, 2010*; *Hjalmarsson et al., 2018*; *Molina et al., 2017*; *Ruiter, Boyle & Zhou, 2013*; *Zhou, Kjer & Morse, 2007*), since larvae often lack significant visible interspecific features (*Johanson, 2007*; *Ruiter, Boyle & Zhou, 2013*; *Zhou, Kjer & Morse, 2007*), and has also turned out to be a valuable tool in stream monitoring routines (*e.g.*, *Behrens-Chapuis, Herder & Geiger, 2021*).

In this study, we assess the Trichoptera diversity of the MHSNP, apply four different molecular taxon delimitation methods (bPTP, AGBD, ASAP, and GMYC) to estimate the number of species entities, and associate larva and adults by DNA barcoding as part of the Indonesian-German IndoBioSys (Indonesian Biodiversity and Information System) project. OTUs are compared with the BOLD database (*Ratnasingham & Hebert, 2007*) to assign species names when possible and to determine the number of putative species missing in genetic reference libraries. Our results contribute to a better understanding of the Trichoptera diversity in West Java.

## MATERIALS AND METHODS

### Taxon sampling

Larval and adult trichopteran specimens were collected at 26 sampling sites between 252 and 1,400 m above sea level in 2015 (dry season, September) and adults additionally in 2016 (wet season, April) (Research permit no. 339/SIP/FRP/E5/Dit.KI/IX/2015 by the Ministry of Research, Technology, and Higher Education of the Republic of Indonesia). Sites inside the MHSNP in West Java, Indonesia, were selected as part of a larger biodiversity assessment study of this area (*De Araujo et al., 2018*). Most of the sampling sites were therefore located in streams and tributaries along a gradient of human impact from agricultural areas to secondary and primary forests. Additionally, samples from the vicinity of Bogor were included. The sampling of the larval Trichoptera followed a multi-habitat sampling approach of 20 pooled sampling units along a 100 m stream stretch with the standard kick-sampling method (*Barbour et al., 1999*) and stone washing from down-to upstream using a dip net (standard Heberle net 25 × 25 cm frame; 2 mm mesh-sized). Adults were collected in the field by sweeping with a net or using a light trap. Specimens were preserved in 96% ethanol. Larval morphospecies were primarily identified based on the taxonomical keys of *de Moor & Ivanov (2008)*, and *Yule & Yong (2004)*. The identification of adult specimens followed *Malicky, Ivanov & Melnitsky (2011)*, *Malicky, Malicky & Chantaramongkol (2010)*, *Ulmer (1913*, *1930*, and *1951)*.

### DNA extraction, amplification, and sequencing

DNA was extracted from 180 specimens using the standardized Glass Fiber Plate DNA Extraction protocol of the Canadian Center for DNA Barcoding (CCDB) (*Ivanova, Dewaard & Hebert, 2006*). PCR followed the CCDB protocol and a 658-bp fragment of the mitochondrial gene COI was amplified using the primer pairs LCO1490-JJ and HCO2198-JJ (*Astrin & Stüben, 2008*). Samples were sequenced at the CCDB.

## Sequence and phylogenetic analysis

The primary assembly and trimming of the raw data followed the protocol of CCDB and were uploaded by them to BOLD. The COI haplotype sequences were combined with two selected lepidopteran outgroup species (*Triodia sylvina*, Accession No.: JN307373 and *Dyseriocrania subpurpurella*, Accession No.: HQ563464). The sequences were aligned using MUSCLE (*Edgar, 2004*) and edited in Geneious (v.7.1.9; Biomatters, Auckland, New Zealand).

COI gene trees were reconstructed using both Bayesian inference (BI) and Maximum Likelihood (ML). In addition to a phylogenetic tree including all sequences, we calculated separately phylogenetic trees for families with more than 10 sequences. For the family trees, we used two more closely related and one distant related sister group. Bayesian trees were inferred using BEAST (v.1.8.3; *Drummond & Rambaut, 2007*), with two independent runs from a random starting tree, an uncorrelated lognormal relaxed clock, and a Yule tree prior. Data were partitioned to codon position and the best substitution model for each partition was selected as part of the analyses using the package bModeltest (v.1.0.4; *Bouckaert & Drummond, 2017*). The Markov Chain Monte Carlo chains (MCMC) ran twice for 30 million generations, with sampling at every 3,000 generations. Convergence was checked using Tracer (v.1.7; *Rambaut et al., 2018*). ML trees were inferred in IQ-TREE (v. 2.1.2; *Nguyen et al., 2014*) with an automated model selection for each partition, consequential GTR as a model for the first codon, and TIM3 and TIM2 for the second and third codon respectively. Branch support was calculated based on 1,000 ultrafast bootstrap replicates.

## Molecular taxon delimitation analysis

Four different molecular taxon delimitation approaches were applied for the complete Trichoptera data set and independently also for Calamoceratidae, Hydropsychidae, Lepidostomatidae, Leptoceridae, Philopotamidae, and Psychomyiidae: the generalized mixed Yule coalescent model (GMYC; *Fujisawa & Barraclough, 2013*; *Pons et al., 2006*), the Bayesian Poisson Tree Processes (bPTP; *Zhang et al., 2013*), the Automatic Barcode Gap Discovery (ABGD; *Puillandre et al., 2012*), and the Assemble Species for Automatic Partitioning (ASAP; *Puillandre, Brouillet & Achaz, 2021*). The ultrametric ML trees used for GMYC were generated using the chronos function in the ape v. 5.2 package (*Paradis & Schliep, 2019*) in R (v3.5.2; *R Core Team, 2018*). Four different clock models were tested: strict, discrete with 10 rate categories, correlated and uncorrelated-relaxed. The best models were selected based on the $\varphi$ information criterion by *Paradis (2013)*, which takes the penalized term into account. All models were fitted on lambda set to 1.0 and in all cases, the strict clock was found to be the best-fitting model. The single threshold versions of GMYC were run on the maximum credibility trees inferred with BEAST and the ultrametric ML trees in R using the package splits (*Ezard, Fujisawa & Barraclough, 2009*). The bPTP analyses were carried out using the bPTP web server (http://species.h-its.org/; *Zhang et al., 2013*) based on the ML trees, with 100,000 MCMC generations, sampling every 100 generations, the burn-in set to 0.1, and including the respective outgroups. For the AGBD analyses, alignments were submitted to the AGBD online web server

(https://bioinfo.mnhn.fr/abi/public/abgd/; *Puillandre et al., 2012*), with P (prior intraspecific divergence) set from 0.001 to 0.1 and steps set to 10, X (minimum relative gap width) set to 1, Nb bins (from distance distribution) set to 20, the Kimura (K80) model substitution model (*Kimura, 1980*) and TS/TV to 2.0. The initial partitions with the lowest prior intraspecific divergence (*P*) value were selected as the best AGBD delimitations. Similarly, for ASAP analyses, the alignments were submitted to the web interface (https://bioinfo.mnhn.fr/abi/public/asap; *Puillandre, Brouillet & Achaz, 2021*) selecting K2P (*Kimura, 1980*) as the substitution model and other parameters left as default. The partitions with the highest *P*-value were selected here as the best ASAP delimitation since the commonly-used asap-score based on the barcode gap is prune to sampling efforts (*Wiemers & Fiedler, 2007*). Moreover, to understand how many of the OTUs are already present in BOLD, sequences were compared against the BOLD database using BLAST (www.boldsystems.org; *Camacho et al., 2009*; *Ratnasingham & Hebert, 2007*). A solid match to a species was assumed when the hit was higher than 99% similarity, at genus level when ≥95%, and family ≥91% as a rough proxy, following *Coddington et al. (2016)* and *Elbrecht et al. (2017)*.

### Life stage associations

Only OTUs for which at least two molecular taxon delimitation approaches yielded the same results (*Carstens et al., 2013*) were used to further investigate and discuss possible associations between larvae and adults.

## RESULTS

### Morphotype grouping and phylogenetic inference

The morphological identification of 180 adult and larval specimens resulted in 65 morphospecies from 15 families. Larval specimens belonged to at least 15 morphospecies from 12 families, whereas a total of 51 species from 12 families could be identified from adult specimens (Table 1). Brachycentridae, Ecnomidae, Dipseudopsidae, and Polycentropodidae were represented only by larvae, while Helicopsychidae, Calamoceratidae, and Xiphoncentridae were represented only by adult specimens. Brachycentridae, Ecnomidae, and Helicopsychidae were only represented by singletons (Table S1).

Sequence length ranged from 363 to 658 bp with at least 36.5% of identical sites and a GC content of 32.7%. The resulting trimmed COI sequences are deposited in GenBank (Accession numbers provided in Table S2). Gene trees based on ML (Fig. S1) and BI trees (Fig. 1) of the COI sequences produced overall similar topologies with just minor differences. Main differences were in the relationships between Ecnomidae-Pseudoneuroclipsidae, Hydroptilidae-Glossosomatidae, and Brachycentridae-Goeridae. Morphospecies formed monophyletic entities in almost all cases in the ML tree, except for *Glossosoma javanicum* (Glossosomatidae), *Ganonema fuscipenne* (Calamoceratidae), and Leptoceridae (9.2.III). In the BI tree, *G. javanicum* was paraphyletic and *Triplectides indicus* (Leptoceridae) was misplaced as the sister group of Calamoceratidae.

**Table 1 Trichoptera diversity in and around the Mount Halimun Salak National Park, West Java, derived from adult and larval records (no. of specimens).**

| Family | Taxa | Larva | Adult |
|---|---|---|---|
| Brachycentridae | | 1 | |
| Calamoceratidae | *Anisocentropus flavomarginatus* Ulmer, 1906 | | 3 |
| Calamoceratidae | *Anisocentropus* sp. | | 1 |
| Calamoceratidae | *Ganonema* sp. | | 1 |
| Calamoceratidae | *Ganonema fuscipenne* Albarda, 1881 | | 4 |
| Calamoceratidae | *Ganonema ochraceellum* McLachlan, 1866 | | 2 |
| Pseudoneureclipsidae | | 2 | |
| Ecnomidae | | 1 | |
| Glossosomatidae | *Glossosoma javanicum* Ulmer, 1930 | | 2 |
| Glossosomatidae | *Glossosoma* sp. | 1 | |
| Goeridae | *Goera conclusa* Ulmer, 1905 | | 2 |
| Goeridae | *Goera* sp. | 1 | |
| Goeridae | | 3 | |
| Helicopsychidae | *Helicopsyche* sp. | | 1 |
| Hydropsychidae | *Agapetus* sp. | 5 | |
| Hydropsychidae | *Cheumatopsyche globosa* Ulmer, 1910 | | 1 |
| Hydropsychidae | *Cheumatopsyche lucida* Ulmer, 1907 | | 2 |
| Hydropsychidae | *Cheumatopsyche* sp. | | 2 |
| Hydropsychidae | *Diplectrona gombak* Olah, 1993 | | 2 |
| Hydropsychidae | *Diplectrona pseudofasciata* Ulmer, 1909 | | 1 |
| Hydropsychidae | *Diplectrona ungaranica* Ulmer, 1951 | | 1 |
| Hydropsychidae | *Hydromanicus flavoguttatus* Albarda, 1881 | | 4 |
| Hydropsychidae | *Hydromanicus* sp. | | 2 |
| Hydropsychidae | *Hydropsyche saranganica* Ulmer, 1951 | | 3 |
| Hydropsychidae | *Hydropsyche* sp. | | 1 |
| Hydropsychidae | | 32 | 2 |
| Hydroptilidae | *Orthotrichia* sp. | 1 | |
| Hydroptilidae | *Scelotrichia* sp. | | 2 |
| Hydroptilidae | | | 1 |
| Lepidostomatidae | *Lepidostoma diehli* Weaver, 1989 | | 2 |
| Lepidostomatidae | *Lepidostoma jacobsoni* Ulmer, 1910 | | 1 |
| Lepidostomatidae | *Lepidostoma longipenis* Weaver, 1989 | | 1 |
| Lepidostomatidae | *Lepidostoma* sp. | | 4 |
| Lepidostomatidae | | 5 | |
| Leptoceridae | *Adicella byblis* Malicky, 1998 | | 1 |
| Leptoceridae | *Adicella pulcherrima* Ulmer, 1906 | | 1 |
| Leptoceridae | *Adicella* sp. | | 4 |
| Leptoceridae | *Oecetis kapaneus* Malicky, 2005 | | 2 |
| Leptoceridae | *Oecetis oviformis* Ulmer, 1951 | | 1 |
| Leptoceridae | *Oecetis* sp. | | 4 |
| Leptoceridae | *Oecetis tripunctata* Fabricius, 1793 | | 4 |

| Table 1 (continued) | | | |
|---|---|---|---|
| **Family** | **Taxa** | **Larva** | **Adult** |
| Leptoceridae | *Setodes larentia* Malicky & Chantaramongkol, 2006 | | 1 |
| Leptoceridae | *Setodes musagetes* Malicky & Chantaramongkol, 2006 | | 2 |
| Leptoceridae | *Setodes* sp. | | 2 |
| Leptoceridae | *Tagalopsyche brunnea* Ulmer, 1905 | | 3 |
| Leptoceridae | *Triplectides indicus* Walker, 1852 | | 1 |
| Leptoceridae | *Trichosetodes handschini* Ulmer, 1951 | | 3 |
| Leptoceridae | | 7 | |
| Philopotamidae | *Chimarra briseis* Malicky, 1998 | | 3 |
| Philopotamidae | *Chimarra* sp. | | 3 |
| Philopotamidae | *Chimarra* sp., cf. *aram* | | 1 |
| Philopotamidae | *Gunugiella* sp. | | 1 |
| Philopotamidae | | 10 | |
| Polycentropodidae | | 3 | |
| Psychomyiidae | *Lype atnia* Malicky & Chantaramongkol, 1993 | | 2 |
| Psychomyiidae | *Paduniella* sp. | | 1 |
| Psychomyiidae | *Psychomyia capillata* Ulmer, 1910 | | 2 |
| Psychomyiidae | *Psychomyia* sp. | | |
| Psychomyiidae | *Tinodes* sp. | | 4 |
| Psychomyiidae | *Tinodes* sp.nov. | | 1 |
| Rhyacophilidae | *Rhyacophila* sp. | | 1 |
| Rhyacophilidae | | 5 | |
| Xiphocentronidae | *Drepanocentron* sp. | | 3 |
| Xiphocentronidae | *Drepanocentron* sp.1 | | 2 |
| Xiphocentronidae | *Drepanocentron* sp.2 | | 1 |
| *Unknown* | | 1 | |
| *Total* | | 78 | 102 |

## Molecular diversity estimation

Species diversity was analyzed based on morphology and four different statistical molecular taxon delimitation approaches: GMYC, bPTP, ABGD, and ASAP. Overall, the total number of species yielded by all statistical methods was very similar with 74, 81, 72, and 72 respectively, representing 15 families in total.

Analysis with ABGD generates two different results: the initial and the recursive partition (see Fig. S2). The recursive partitions usually generate a higher number of clusters; however, the initial partition has proven to give results that better match group assignments of expert taxonomists (*Puillandre et al., 2012*). Recursive partitions of the full data set ($n$ = 128 haplotypes) showed that the group assignments range from 1 ($P$ = 0.1) to 76 ($P$ = 0.001). On the other side, initial partitions resulted in 72 groups with $P$-values between 0.001 and 0.1000. ASAP similarly yielded 72 putative species (Fig. S3). The GMYC analysis on the BI tree delimited 74 OTUs in total (Fig. S4), and the bPTP method recovered 81 OTUs (confidence interval between 57 to 96 species). GMYC analysis of the

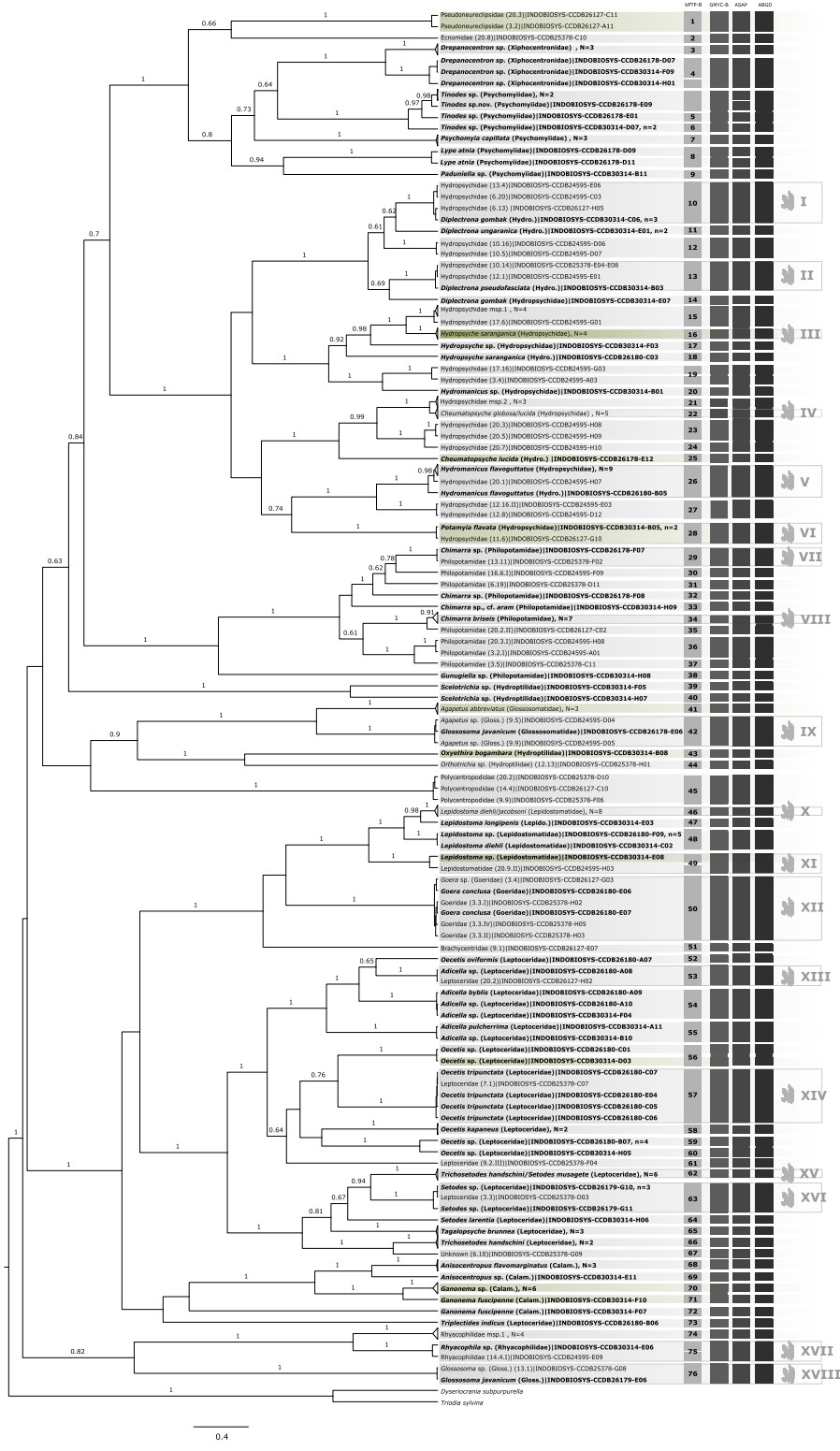

**Figure 1 Bayesian inferred ultrametric phylogenetic tree based on COI of 182 Trichoptera larvae (in bold) and adults.** Bootstrap values higher than 0.5 are indicated above branches. Results of the four different molecular taxon delimitation approaches (bPTP, GMYC, ASAP, and AGBD) are illustrated by vertical bars. Each bar represents an Operational Taxonomic Unit (OTU) detected by the respective

**Figure 1 (continued)**
approach labeled on the top. Morphological identification of adult (in bold) and juvenile specimens were added directly to the leaves. Green tips are species also identified in BOLD following the thresholds by *Elbrecht et al. (2017)*. OTUs endorsed by at least two of the four delimitation methods are numbered in the first column and BLAST results are summarized in Table 1. Additionally, confirmed OTUs with juvenile and adult specimens are marked with a symbol at the end of the vertical bars. *N* = number of OTUs collapsed in the phylogenetic tree, *n* = number of haplotypes collapsed to one sequence.

**Table 2 Total number of OTUs per family based on the different molecular taxon delimitation approaches.**

| Family | GMYC-B | bPTP-B | ABGD | ASAP |
|---|---|---|---|---|
| Brachycentridae | 1 | 1 | 1 | 1 |
| Calamoceratidae* | 4 | 5 | 5 | 5 |
| Pseudoneureclipsidae | 1 | 1 | 1 | 1 |
| Ecnomidae | 1 | 1 | 1 | 1 |
| Glossosomatidae | 3 | 3 | 3 | 3 |
| Goeridae | 1 | 1 | 1 | 1 |
| Hydropsychidae* | 19 | 20 | 17 | 14 |
| Hydroptilidae | 4 | 4 | 4 | 4 |
| Lepidostomatidae* | 4 | 5 | 4 | 4 |
| Leptoceridae* | 16 | 16 | 15 | 15 |
| Polycentropodidae | 1 | 1 | 1 | 1 |
| Philopotamidae* | 10 | 10 | 10 | 9 |
| Psychomyiidae* | 4 | 6 | 4 | 8 |
| Rhyacophilidae | 2 | 3 | 2 | 2 |
| Xiphocentronidae | 2 | 3 | 2 | 2 |
| *Unknown* | 1 | 1 | 1 | 1 |
| *Total* | 74 | 81 | 72 | 72 |

**Note:**
* Families analysed separately with two more closely related and one distant related sister group.

ML tree yielded 92 OTUS (confidence interval of 90 to 109 species, and 18 more than on the BI tree) (see Fig. S1), whereas bPTP on the ML tree yielded 89 OTUs (confidence interval between 75 to 116 species and seven more than on the BI tree).

When applying these four molecular taxon delimitation methods to our data, main discrepancies in the number of OTUs were found within Hydropsychidae, Lepidostomatidae, Leptoceridae, Philopotamidae, Psychomyiidae, Rhyacophilidae, and Xiphocentridae (Table 2). Dissimilarities were caused mainly by AGBD and ASAP being more conservative in comparison with bPTP and GMYC. Of all families present, Hydropsychidae had an overall much higher richness with 21 OTUs. Five families were only represented by a single OTU each.

A total of 44 of 60 OTUs delimited unanimously with three molecular taxon delimitation approaches could be identified to family level in the BOLD database. Only five OTUs could be identified to species level and four to genus level. In 10 cases the OTUs

could only be identified as Trichoptera (see green labels in Fig. 1) when querying it to the reference library of BOLD. Overall, the morphological identification was consistent with the molecular species diversity delimitation, but there were five exceptions: *Hydropsyche saranganica* (Hydropsychidae) and *Ganonema fuscipenne* (Calamoceratidae) were divided into two OTUs, *Glossosoma javanicum* formed a cluster with *Agapetus* sp. (Glossosomatidae), and *Lepidostoma diehli* and *L. jacobsoni*, and *Trichosetodes handschini* and *Setodes musagete*, respectively, formed a single OTU (Lepidostomatidae).

### Larval-adult association

In 18 of the 60 consensus OTUs, representing six families, an association of larval and adult stages was possible (see Table S1). In five cases, the association involved two or more specimens for each life stage. In 13 cases, either the larval or adult stage was represented by only one specimen. It was possible to associate 33 larvae and 35 adults of the following taxa: *Diplectrona gombak*, *Diplectrona pseudofasciata*, *Hydropsyche saranganica*, *Hydromanicus flavoguttatus*, *Potamyia flavata*, *Chimarra* sp., *Chimarra briseis*, *Goera conclusa*, *Adicella* sp., *Oecetis tripunctata*, *Setodes* sp., and *Rhyacophila* sp. *Agapetus* sp./*Glossosoma javanicum*-complex, *Lepidostoma diehli/jacobsoni*-complex, *Cheumatopsyche globosa/lucida*-complex, and *Trichosetodes handschini/Setodes musagetes*-complex.

## DISCUSSION

### Phylogenetic reconstruction and delimitation methods

The gene trees inferred based on the DNA-Barcode COI fragment of 180 Trichoptera larvae and adults with ML and BI, show very similar topologies. In general, the structure and position of superfamilies and families of both trees reflect the current phylogeny of Trichoptera (*Thomas et al., 2020*). The main incongruence between ML and BI trees is the unclear relationships between two genera within Glossosomatidae, *Glossosoma*, and *Agapetus*. COI data suggest that both genera are polyphyletic, which is the case for multiple genera within Trichoptera (*de Moor & Ivanov, 2008*). However, phylogenetic inferences based on single and fast-evolving mitochondrial markers have to be treated with caution, also in Trichoptera. The main goal in larval associations is to find the closest match between adult and larval specimens, in this case with a molecular taxon delimitation approach on a COI tree (*Zhou, Kjer & Morse, 2007*).

### Species delineation and the overall Trichoptera diversity of West Java

To our knowledge, this is the first molecular assessment of the Trichoptera diversity of West Java. A total of 72 to 81 OTUs within 15 Trichoptera families were recovered. In three of the species identified morphologically, the molecular taxon delimitation methods unanimously split the species into two different OTUs. The only previous study of the Trichoptera of this area reported 24 species in 12 families near rice fields in the MHSNP (*Rizali, Buchori & Triwidodo, 2002*); the exact information on which species were recorded is however not available. Nonetheless, we can confirm the presence of twelve of the sixteen families reported by *Rizali, Buchori & Triwidodo (2002)*. Moreover, species of

six additional families (Calamoceratidae, Dipseudopsidae, Ecnomidae, Goeridae, Lepidostomatidae, Psychomyiidae) are identified. Different sampling methods and the incorporation of different habitats likely explain the discrepancy. Although already discovering much higher diversity than in the previous study, the number of species reported here is likely only a fraction of the diversity in the study area.

A comprehensive study on Indonesian Trichoptera diversity (*Malicky, Ivanov & Melnitsky, 2014*) reported 146 species from 16 families, based on collections and published data. Except for Stenopsychidae, all of the families recorded by *Malicky, Ivanov & Melnitsky (2014)* are also reported here, from one single national park. Moreover, several species recorded here are first records for Java or even undescribed (Mey et al. in preparation). In the families Brachycentridae, Lepidostomatidae, and Calamoceratidae, nearly all species previously known from Java could be confirmed. The diversity of Hydroptilidae was relatively low compared to previous studies. This is likely due to the net size (2 mm) applied here, where the small-sized members of this family might not get collected.

The high number of caddisfly species in this part of West Java suggests the presence of a large range of microhabitats (*Dudgeon, 2011*). However, due to illegal gold mining and settlements, this habitat diversity is under threat, also within the park. High concentrations and bioaccumulation of heavy metals (*e.g.*, mercury) have already been found in water and sediments (*Sudarso et al., 2013*; *Sudarso, Wardiatno & Sualia, 2008*). These may cause shifts in Trichoptera diversity (*Loayza-Muro et al., 2010*; *Wiederholm, 1984*), and can lead to morphological abnormalities at least in some species (*Yoga et al., 2014a*, *2014b*).

The application of DNA barcoding is an efficient method to assess Trichoptera diversity in areas with insufficient or even missing taxonomic knowledge, and can help to assess freshwater stream quality (*Sweeney et al., 2011*). This is especially valuable, as a clear morphological identification of caddisfly larvae is in many cases hindered by the absence of reliable morphological characters or lack of publications on the subject (*Hjalmarsson et al., 2018*). Merely 8.6% of all OTUs identified in this study could be identified to species level when only compared against the BOLD database. Therefore, even though we now have a better idea of the Trichoptera diversity present in the national park, their names and thus the ecological features associated with them remain largely unknown. This underlines the need to expand genetic and morphological studies on Trichoptera in poorly known tropical areas with high diversity, to make DNA barcoding a more accessible monitoring method of Trichoptera diversity in the Oriental Region (*Zhou et al., 2016*). If future studies can link the presence of certain species in this area to environmental factors, it would be possible to assess the quality of streams in this area.

## Life stage association with DNA barcoding

DNA barcoding enabled matching caddisfly larvae and adults in 18 cases. In some of these, only a single individual of one life stage (adults or larvae) was available for inferring the association. Of these successfully associated species, only the larva of *O. tripunctata*, a cosmopolitan species (*Waringer & Graf, 2014*), and *G. conclusa* (*Ulmer, 1913*) have been described. Incorporation of higher numbers of both life stages, from a larger number of

sites and covering all relevant habitats, would be required for filling the substantial gap remaining. Moreover, a sound knowledge of both life stages is in turn essential for understanding inter- and intraspecific variation, ontological variations, and finally the species-specific ecology (*Hjalmarsson et al., 2018*). While the present study has once more demonstrated the potential of DNA barcoding to understand and link the life stages of Trichoptera, it also highlights the need for a more complete barcode reference library for this region. The associated taxa could in most cases be identified to species level as a result of the morphological identification of adult specimens and not by larvae or a match in BOLD. Of the associated species represented by an adult and larval specimen in this study, *Potamyia flavata* was the only species present in BOLD. The collected adult and larval specimens with their corresponding DNA barcodes provide valuable information for future studies in this area.

## CONCLUSIONS

The present study highlights the need to further explore the genetic diversity of Trichoptera in tropical areas. The results show the poor state of genetic and morphological exploration of Indonesian Trichoptera, especially the larval stages, which currently hinders further use as bioindicators for freshwater habitat quality in the area. Monitoring freshwater quality would be especially important in the study area since it serves as the major water reservoir for the megacity of Jakarta. Moreover, our results show that the inventory of larval and genetic Trichoptera diversity on Java is far from complete, and substantial gaps remain in linking the OTUs uncovered here to species entities. Likewise, substantial work remains to be done to link trichopteran life stages. Nevertheless, we would argue that upscaling our DNA barcoding approach, further morphological studies of adults and larvae, and progress in Trichoptera taxonomy would represent a decisive move towards translating biodiversity data into a wide applicable monitoring tool.

## ACKNOWLEDGEMENTS

We thank Witjaksono, the Head of Research Center for Biology LIPI, presently Research Center for Biosystematics and Evolution, National Research and Innovation Agency (Badan Riset dan Inovasi Nasional, BRIN) for his support. Thanks also to the late Tri Siswo Rahardjo, the Head of Taman Nasional Gunung Halimun Salak, Ministry of Environment and Forestry for providing support to conduct research in the conservation area under his care. Bruno Cancian de Araujo and Jérôme Morinière of the Zoologische Staatssammlung München (ZSM) were responsible for DNA extractions.

### Funding

The fieldwork and morphological identification were supported by the German Federal Ministry of Education and Research (BMBF; INDOBIOSYS MfN Berlin, 16GW0111K) and by Indonesian DIPA (079.01.2.017148 2015, project numbers 3400.003.050.I and 079.01.2.017148 2016, project number 3400.010.005.061B). The DNA-barcoding was

supported as well by the BMBF (INDOBIOSYS ZSM Munich, 16GW0112). The publication of this article was funded by the Open Access Fund of the Leibniz Association. The funders had no role in study design, data collection and analysis, decision to publish, or preparation of the manuscript.

### Grant Disclosures
The following grant information was disclosed by the authors:
German Federal Ministry of Education and Research.
Indonesian DIPA.
BMBF.
Open Access Fund of the Leibniz Association.

### Competing Interests
The authors declare that they have no competing interests.

### Author Contributions
- Isabel C. Kilian conceived and designed the experiments, performed the experiments, analyzed the data, prepared figures and/or tables, authored or reviewed drafts of the article, and approved the final draft.
- Marianne Espeland analyzed the data, prepared figures and/or tables, authored or reviewed drafts of the article, and approved the final draft.
- Wolfram Mey performed the experiments, authored or reviewed drafts of the article, and approved the final draft.
- Daisy Wowor conceived and designed the experiments, authored or reviewed drafts of the article, and approved the final draft.
- Renny K. Hadiaty conceived and designed the experiments, authored or reviewed drafts of the article, and approved the final draft.
- Thomas von Rintelen conceived and designed the experiments, authored or reviewed drafts of the article, and approved the final draft.
- Fabian Herder conceived and designed the experiments, authored or reviewed drafts of the article, and approved the final draft.

### Field Study Permissions
The following information was supplied relating to field study approvals (*i.e.*, approving body and any reference numbers):

Research permit was generated by the Ministry of Research and Higher Education of the Republic of Indonesia (RISTEKDIKTI) (permit no. 339/SIP/FRP/E5/Dit.KI/IX/2015).

### Data Availability
The raw metadata is available in the Supplemental File and the 180 sequences are available in BOLD (Table S2).

## Supplemental Information

Supplemental information for this article can be found online at http://dx.doi.org/10.7717/peerj.14182#supplemental-information.

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
