# Peer review of "DNA barcoding unveils a high diversity of caddisflies (Trichoptera) in the Mount Halimun Salak National Park (West Java; Indonesia)"

_PeerJ, doi:10.7717/peerj.14182_

## Round 0.1 · original submission · Minor Revisions

I have now received two reviewer reports for your study, both of which are constructive in nature that will help enhance your study. They feel the study is useful, the data is extensive but feel that the presentation and interpretation of the results could be improved. I endorse this view. Please address all of their comments and provide a point by point rebuttal.

I draw your attention to two main points. Though attractive as a focal taxon for biomonitoring, the paper in itself does not provide the background to do this, so consider toning down on this context. Furthermore, the unidentified OTUs does not necessarily mean that they are new species, but there is a reasonable possibility that they are, given the patterns of endemicity for Trichoptera in general.

·

Basic reporting

No comment.

Experimental design

Original primary research within Aims and Scope of the journal.
Yes

Research question well defined, relevant & meaningful. It is stated how research fills an identified knowledge gap.


Rigorous investigation performed to a high technical & ethical standard.
0 - see general comments.


Methods described with sufficient detail & information to replicate.
1 – But provide information how substitution models for BI were selected. New versions of BEAST will do this for you. Also provide information if and how data were partitioned, and if data were not partitioned, explain why.

Validity of the findings

I am afraid that I feel a bit uneasy with some parts of the manuscript, summarized in the general comments.
Generally, I am not sure if this work really is conducive to biomonitoring because: no specific sampling strategy was employed that could have linked certain taxa to human impact.
The conclusion that the Trichoptera fauna of Java is underestimated is valid – simply because it's a large island and there was limited sampling effort – but does not necessarily stem from the data presented here.
Life stage association is an excellent tool to furnish taxonomic keys to larval stages, but only if there is species-level resolution.
Finally I must point out that all methods used here were shown to poorly estimate Trichoptera diversity at species-level (see Vitecek et al. 2017, BMC Evolutionary Biology). I do not request that this paper should be cited, but highly recommend its perusal. There are quite some things to consider when attempting molecular species hypotheses inference and, like everything in taxonomy, this should be done properly or not at all.

Additional comments

Review Kilian et al. PeerJ-60931

Comments en gros
This is another barcoding paper, using a well-established method to assess biodiversity in an area where no man or woman has barcoded before. I am delighted to see that the authors pursue other delimitation approaches and do not follow the BIN-theory.
From the abstract I would have expected some morphological study – what use is the affiliation of larvae and adults if the larvae are not described? – and from the title a much greater overall sampling size and a much more taxonomic focus.
And while I agree with the authors conclusions, there still remains the issue of the taxonomic impediment so that I am left wondering if and how DNA barcode reference libraries can be improved if there is no one to do the work.

Comments en detail
Introduction
48:50 There are better references than this. Consider Graf et al., Schmidt-Kloiber and Hering, etc. Also refer to freshwaterecology.info
50:52 This is wrong. Physicochemical approaches provide different information, compared to ecological monitoring – and while they may be less expensive, good ecological monitoring comes with a plethora of prerequisites including a certain taxonomic prowess.
70:73 Rizali et al. 2002 as cited did not focus on aquatic insect diversity, and certainly did not detect 269 species of caddisfly. The corresponding paper reports a total number of 435 species, with 24 Trichoptera species in total (if my maths are correct) – I assume the authors misread the table (Tabel 4. Jumlah individu (N) dan spesies (S) tiap ordo dan famili serangga yang ditemukan pada tiap perangkap; Individual (N) and species (S) numbers per insect order and family collected by each trap type). It appears that you took the wrong column. Rizali et al. explored biodiversity at forest-agriculture ecotones using 4 different trap types, which of course are heavily disturbed. Resolve and cite properly.
74:85 I assume you mean INTERspecific variability? INTRAspecific variability is a given, but taxonomy usually aims at discerning INTRA- from INTERspecific variation. Concerning the «valuable standard tool» statement it will be necessary to point out that not everyone shares this point of view, and there are quite strong opinions on the inadequacy of barcoding as biodiversity exploration tool.
88 Again, what is the benefit of associating larvae and adults if there are no practical consequences?
89:90 Did you also try to identify the material using the standard approach?

Materials and Methods
Taxon sampling should be described in greater depth, and especially the sampling sites should be described.
105 2 mm mesh size? Really? Could explain the scarcity of larvae. Hydroptilids will likely not be collected with such a net.
107:108 Malicky et al. do not consider larvae and certainly do not present a key to larvae – how could this have been used for larval identification? Side note: Hans Malicky disdains larvae (unless they can be reared to adults) and remains convinced that all larval-based assessments are dubious to say the least.
124 How where substitution model selected? Were data partitioned? Did you use different models for different partitions? This question pertains to both ML and BI analyses.
125:127 Which ploidy settings where used, and why?
143:145 Did you consider the alternatives (confidence interval, etc.) provided by GMYC? Do the species hypotheses inferred on the Ml tree and the BI tree differ? You report differences in their topology, did this affect species hypotheses inference? Provide all diagnostic plots in the supplement.
145:148 I can't recall if bPTP provides a confidence interval to the inferred species as well. If this is available, provide this information.
148:153 Concerning ABGD: Did you test different Ps, Xs and Nb bins? How where these values selected? Why select the K2P substitution model? How where results, i.e., species hypotheses selected? Why did you use (outdated) ABGD instead of ASAP (https://bioinfo.mnhn.fr/abi/public/asap/asapweb.html#), a thoroughly improved version of ABGD? Provide all diagnostic plots in the supplement.
156:158 This seems a little odd. Surely if a set of larvae and adults are recovered as the same OTU by the three molecular species hypotheses inference methods they can be considered conspecific and thus associated?
160:162 I have had numerous discussions now about how families and genera should be identified using BOLD, or any other reference library. The problem is that genera are artificial constructs that group species often based on the principle of «Don't know where that thing belongs, so I' going to put it into genus XXX» or «Better describe a new genus for that one» – this results in relatively poor systematics, and a poor base for genus-assignment based on BOLD (not touching upon the far greater issues of database quality, curation and completeness). My approach would have been to download BOLD sequence data for the genera in question, run IQtree (for speed) with these and my query sequences and use the output to assess where the taxa I have are located in the tree.
177:183 How high is the probability of contamination of samples that could have led to the observed patterns?


Generally, I am wondering if an how these molecular delimitation tools can be used on such diverse datasets as presented here. If species in different clades are of different age, both tree-based and distance-based approaches will not recover meaningful results for groups comprising older-than-average or younger-than-average species. I think this needs to be discussed and considered.

Results
187:189 Again, is there a difference between GMYC results of BI and ML trees?
206:209 Could this be a result of differences in age between clades and species?
215:220 While the species-level assignments are relevant and important, I am at a loss concerning the others. If species-level assignment cannot be achieved, how can species-level larval keys be developed? How could these results be presented in a clearer manner? Present the most important (i.e. species-level) results first, and then the others.

Discussion
224 You have only partial COI data.
228:229 Could this also be a contamination issue? Or a sequence length issue? Otherwise, it's quite conceivable that these are poorly defined genera – see my above comment.
237-238 I fail to follow the authors argument concerning the number of OTUs with matching morphological identification; there are 5 species-level OTU identifications as reported in the results section. Are you referring to family/genus-level identifications? Personally, I would refrain from reporting these as significant results. Family/genus identification should be feasible with very limited effort based on morphological assessments alone.
238:240 This is again the poor source work from lines 70:73. Rizali et al. did not recover 269 species of Trichoptera but 269 specimens, and 24 Trichoptera species.
254:255 Yes, but a detailed description of sampling sites could provide relevant information as concerns which habitat types there might have been. I went to Taman Nasional Gunung Halimun Salak, and I remember few aquatic habitats (focussing on butterflies in the framework of a joint field trip of the University of Vienna and the Insitut Pertanian Bogor University), but there must have been some criteria based on which the sampling sites were selected.
275:281 Yes, but to what level of identification? A larva associated to a species complex is not useful. If there is species-level assignment but no morphological differences can be found to discern larvae that's one thing. The problem of resolving autecology can only addressed when larvae are known, or at least sufficiently well-known to explore ecological niche space.
286 Potamyia

Conclusions
291:292 Well, not really. The majority of species were identified based on classical taxonomy approaches, not by DNA-barcoding. Rephrase.
294:295 Sentence incomplete.
297:298 How do your results show that the Trichoptera inventory is far from complete? It's to be expected that there are more than the 140-odd species of Malicky et al., but this is by no means demonstrated here. The fact that you found inassignable OTUs can be parsimoniously explained by the lack of barcoding effort. That not all specimens could be identified to species level must not necessarily mean that these are new species. And if they are I would rather see the species descriptions in the same manuscript as the molecular species hypotheses inference. There is no need to separate these parts.
300:302 True, but only if there is a proper calibration framework for monitoring tools.

How can the data generated here be used in the framework of biomonitoring? Biomonitoring needs reference conditions, and gradients of human impact – if there is proper site selection then this may be true, but to extrapolate relevance of these data for biomonitoring from the general principles is exaggeration.

References
Italicise species names throughout.

Figures
The tree should be provided in scalable vector format; the current form is almost indecipherable.

Supplement
Provide all relevant diagnostic plots and label them.

·

Basic reporting

No comment.

Experimental design

As usual in DNA analyses, authors used several software to edit and analyze the sequences. However, it is unclear the exact use of some of these software, for example, it is not clear when authors use Sequencher or Geneious for sequence editing.

Validity of the findings

No comment.

Additional comments

The manuscript provides an important contribution to the knowledge of caddisfly. Although the molecular taxonomy has been successfully used for a large variety of organisms, the data available for Trichoptera are still very scarce. I have only a few suggestions and comments, which I have included in the PDF file. I recommend the manuscript to be accepted to publication in PeerJ, with minor revisions.

---

## Round 0.2 · Minor Revisions

One of the reviewers has concerns about the context of the species delimitation analyses performed. I agree with the reviewer that widely disparate clades should not be considered in a single analysis. Since the analyses outlined will improve your species delimitation outcomes, and given that running these analyses is straightforward, I urge the authors to consider carrying out these analyses and responding to the reviewer's comments.

·

Basic reporting

Only marginal comments. The authors successfully improved their manuscript. I am afraid to state at this stage that a little language shine would do well.
The term «species delimitation» should be avoided. From the way the authors write their manuscript, they are well aware of the issues associated with these tools, and they use them only to infer OTUs – some comments below address this point – and this should be resolved and reflected in how these tools are referred to.
Other than that, no comments.

Experimental design

I have 2 comments:

1. In response to a comment on how well these molecular delimitation tools can be used on a diverse dataset, comprising many lineages (and therefore species) of different evolutionary age, the authors respond the following:

[Original reviewer comment: Generally, I am wondering if an how these molecular delimitation tools can be used on such diverse datasets as presented here. If species in different clades are of different age, both tree-based and distance-based approaches will not recover meaningful results for groups comprising older-than-average or younger-than-average species. I think this needs to be discussed and considered.]

[Author response] Species and clades in general are always of different age independently of the size of the analysed dataset. Therefore, we will never be able to avoid this problem in general. Nonetheless, because we have a diverse dataset, the numerous delimitation approaches we used are the best options we have. If all species would be of similar age we would likely not need to use statistical methods for species delimitation in the first place.

To me, this response is unfortunately not satisfying. The most recent (and to my understanding most adequate) species concepts all place an emphasis on independent evolution of a lineage to become a species – this means that a threshold time of independent evolution (i.e., without gene flow between lineages) must be passed before an entity can be considered a species. All molecular delimitation methods used in the present manuscript aim to somehow detect that threshold, and to distinguish intra-"specific" variation from inter-"specific" variation. A taxonomist does the same, but with different characters.

So, I played with your dataset – IQ-Tree to infer a tree – and re-ran bPTP via the online tool and got the same results as you reported. Then, as I was wondering if there would be effects of the "neighbourhood" on numbers of inferred entities, I separated the original alignment into families (morphology but also BLASTing would easily make that possible) + outgroup, inferred separate trees for each family where taxon and specimen numbers where high enough, and re-ran bPTP as specified in the manuscript (i.e., naming the outgroups but not removing them).
Doing that, I get 4 Calamoceratidae, 20 Hydropsychidae, 17 Leptoceridae and 4 Psychomyiidae – nothing exciting, but a bit different from your figures.

Then (being lazy and not in the mood to make all these ML trees ultrametric, and not wanting to set up separate BEAST runs for all of these) I went to ABGD and ASAP and ran the alignments, as described (both without the outgroup data). Again, I get 4 Calamoceratidae, but (fasten seatbelts) 14 [ASAP, selecting the best partition based on p-value – not the one with the best ASAP score because I don't understand why a narrower barcoding gap width should be indicating the quality of a partition; this maybe is something that can be observed with larger specimen numbers and greater taxon sampling; the barcoding gap approach itself has quite some issues with sampling effort – see Wiemers & Fiedler 2007] or 17 [ABGD, but only when setting the barcoding gap width prior X to 0.5-1.0 and that doesn't really make sense to me either; this essentially means that here we're dealing with a taxon sampling where barcoding gap width is very low, which possibly indicates you're dealing with some rather diverse taxa if "intraspecific" distances are high, or where "interspecific" distances vary within the group – but looking at the graph I found that the barcoding gap approach is not really suited for your sample, because you have several groups of distances, so poor taxon sampling also may be a problem here] Hydropsychidae, 14 [ASAP, most significant partition] or 15 Leptoceridae [AGBD, but the same pattern as in Hydropsychidae: several groups of distances – probably again related to poor taxon sampling], and 9 [ASAP, best ASAP score] or 5 [ASAP, hen picking the partitioning result where the threshold falls directly into the barcoding gap as indicated by an almost vertical distance~rank line in the plot (it appears that this method has some issues, too)] or 5 [ABGD] Psychomyiidae.

These results at least partly deviate from those presented by the authors. So, I think there is some sense in analysing the families separately and I would argue that this is the better approach – I do not agree with the authors that the approach of lumping everything into one alignment and analysing that together with mulitple tools is the best approach. The problem here is that each of the method tries to define a single general threshold over clades and species of different age. This simply may not be appropriate. If you're dealing with a set of young species, they'll be lumped together to a single molecularly delimited entity when analysed in batch with a set of old species. Knowing that the different families are of different age (see the Thomas et al. paper) one should account for that when running molecular diversity estimation. I don't think that anyone would lump mammal or bird families together for that purpopse, and it's probably better not to do that when dealing with insects, too.
Properly dealing with the outgroups in the molecular delimitation exercises will also be necessary. It could well be that for family-specific delimitation simply other Trichoptera families may be better outgroups than 2 random (?) Lepidoptera.
My suggestion is that the authors give this per-family analysis a try for those families where they have sufficiently high taxa and specimen numbers. This could provide a more conservative estimate in some cases, and would set a good example (at least in my logic).


2. For ASAP and ABGD describe how you selected the partitioning results.

Validity of the findings

No comment.

Additional comments

Some linguistic comments and other things below.

Line numbers refer to the track-changes manuscript, to make things easier for the authors.
line 28 we carried out a mol.-morph. study ON Trichoptera diversity USING LARVAE FROM A BENTHIC SURVEY AND ADULTS FROM HAND NETTING OR LIGHT TRAPPING. reads smoother like that.
line 49 HABITAT REQUIREMENTS OF CADDISFLY LARVAE
line 53 Unclear what IT refers to. Maybe better spell out what IT's supposed to mean....
lines 57:59 Well, no. The trouble about not knowing species does not impede using certain taxa as bioindicators. This is a common fallacy. Anything can be used as a bioindicator, provided it can be reliably identified and has indicative value (i.e., occurs only/mostly in specific habitats corresponding to near-natural, minimally impaired, impacted, severly modified, or completely messed up habitat classes). Of course it would be best to have the full species set with ecological niches/traits and everything, but if that is not available there are other means – see results of Assess-hkh for an example. Resolve.
lines 60:61 Commas look a bit like belonging to a German sentence; I do that as well and always need to check thrice... Resolve.
lines 75:76 The segue from caddisflies to dna barcoding is a bit rough. Maybe a sequence like «Generally, there are only few current studies on the Indonesian caddisfly fauna. Ecological as well as molecular baseline data are largely missing, and this also pertains to DNA barcoding efforts.» or so may do the trick....
line 107 I think there's again a comma that deserves some attention.
lines 108:111 I understand that the full MHS samples are not dealt with here, but some basic statistics could/should be provided. Like, how many Trichoptera specimens on average per MHS, average total densities per squaremeter, average number of taxa per site, etc. Rough figures like that so that people who are more into this field can get an idea on what the sampling sites were like. A table with these parameters for each sampling site in the supplemental would be grand but is out of scope and therefor not officially requested here.
lines 126:127 So all sequences are already in BOLD? Then please correct misspelled names like «Echomidae» (that's from the alignments in the supplemental).
lines 129:130 species names in italics
line 141 reference to IQ-Tree seemingly missing? there's an author comment there. When using IQ-Tree, it would be nice to know where the substitution models for each partition came from – AUTO mode on a partitioned dataset?
line 149 I would, overall, prefer the use of "species hypotheses inference" rather than "species delimitation analysis". The reason for this is simple: the tools used here can be of help in identifying a new species (by presenting a useful species hypothesis), but they're not delimiting "species" per se. This is just a matter of wording, but I think one should avoid giving the impression that molecular data and computers alone can make taxonomic decisions. Or simply switch to "Molecular taxon delimitation analysis".
line 162 no abbreviations at the beginning of sentences.
line 175 I agree with author M.E. If 2 method agree on an association it's worth considering.
lines 177:181 This is something entirely different from associating life stages. Maybe move to a new section? Could be placed higher up in the text, e.g. at aroung 117:122....
line 187 "up to" may be the wrong term here. "a total" may be better.
line 205 I would decidedly prefer «Molecular diversity estimation» instead of «Estimation of species diversity» – see above comment on molecular species delimitation.
line 221 again the molecular delimitation issue – resolve.
line 227 Are you sure that what was delimited there are species? If not, rephrase. Simply deleting "species" from that sentence would do the trick. Just refer to OTUs (referring to OTUs only could also be a nice way to resolve this «species delimitation issue» – for line 149 that could make "Molecular Operational Taxonomic Unit delimitation" which sounds quite nice, actually)
line 229 4 approaches, for naming these tools find a solution and harmonize throughout the manuscript.
lines 229:232 I don't understand this section. Do you refer to BLASTing your sequences against BOLD? If yes, then please present that properly. Lines 231:232 are particularly confusing in that respect.
In this paragraph, maybe use some of the cool DNA barcoding buzzwords like «DNA barcoding reference library» or so....
lines 232:234 You completely lost me here – what do you mean here? That your morphospecies correspond to the entities inferred by these molecular diversity estimation tools?
lines 234:236 The Code allows for abbreviating genus names with 2 or 3 letters if this is necessary to avoid confusion between species – you could go for Ga. for Ganonema and Gl. for Glossosoma because right now that sentence reads as if there was a species Ganonema javanicum, and this is bad. Better placement of the parenthesis with Lepidostomatidae is necessary – looks now as if referring to Lepidostoma as well as the leptocerids. Again, this is bad. Refer to the families throughout. The current version makes no sense.
line 241 remove "of Trichoptera". no need for that.
line 245 You did not associate larvae and adults to a taxon. You associated larval stages to a taxon, represented by adult specimens. Rephrase and resolve. This is poorly phrased as it is now.
line 257 insert "fragment" after COI, otherwise it makes no real sense.
line 262 a couple of lines above they were paraphyletic. Which? Resolve.
line 261:263 Or maybe the morphological assessment was not entirely correct?
line 273 See comments above – these are not necessarily species. And be careful when doing that with just 1 marker and a few specimens of each species. It may well be that you under-/over-estimate thresholds due to poor taxon sampling and limited number of specimens.
line 306 How do you have a better idea on the number of species? Better call that "Trichoptera diversity".
lines 311:312 this sentence is very long and could be split up, or at least reduced.
line 312 the singular of taxa is taxon.
lines 319:323 Please break that up to multiple sentences!
lines 346:350 Again a candidate for some sentence splitting.

·

Basic reporting

No comment.

Experimental design

No comment.

Validity of the findings

No comment.

Additional comments

The authors have reviewed the manuscript, taking into account all suggestions and comments made in a previous version. The manuscript is clear, well-written, and has been significantly improved. Therefore, based on this new version, I consider it should be published in PeerJ.

---

## Round 0.3 · accepted · Accept

You've addressed the editorial concerns to a substantial extent, and the paper is now much improved. Congratulations on this excellent contribution, and I wish you all the best in your future research.